# Belt Electrode-Skeletal Muscle Electrical Stimulation Prevents Muscle Atrophy in the Soleus of Collagen-Induced Arthritis Rats

**DOI:** 10.3390/ijms26073294

**Published:** 2025-04-02

**Authors:** Kazufumi Hisamoto, Shogo Toyama, Naoki Okubo, Yoichiro Kamada, Shuji Nakagawa, Yuji Arai, Atsuo Inoue, Osam Mazda, Kenji Takahashi

**Affiliations:** 1Department of Orthopaedics, Graduate School of Medical Science, Kyoto Prefectural University of Medicine, Kawaramachi-Hirokoji, Kamigyo-ku, Kyoto 602-8566, Japan; hisamoto@koto.kpu-m.ac.jp (K.H.); a-inoue@koto.kpu-m.ac.jp (A.I.); kenji-am@koto.kpu-m.ac.jp (K.T.); 2Department of Sports and Para-Sports Medicine, Graduate School of Medical Science, Kyoto Prefectural University of Medicine, Kawaramachi-Hirokoji, Kamigyo-ku, Kyoto 602-8566, Japan; 3Department of Immunology, Graduate School of Medical Science, Kyoto Prefectural University of Medicine, Kawaramachi-Hirokoji, Kyoto 602-8566, Japan

**Keywords:** belt electrode-skeletal muscle electrical stimulation, collagen-induced arthritis rat, muscle atrophy, muscle subtype, rheumatoid arthritis

## Abstract

We investigated the effects of belt electrode-skeletal muscle electrical stimulation (B-SES) on muscle atrophy in collagen-induced arthritis (CIA) rats. Twenty-eight 8-week-old male Dark Agouti rats were immunized with type II collagen and Freund’s incomplete adjuvant (day 0). From days 14 to 28, 18 rats received B-SES (50 Hz) four times only on the right hindlimb (STIM), while the contralateral left hindlimb remained unstimulated. Both hindlimbs of 10 untreated CIA rats were defined as controls (CONT). Paw volume was measured every other day. On day 28, the muscle weight, histology, and gene expression of the soleus and extensor digitorum longus (EDL) were analyzed. B-SES did not worsen paw volume throughout the experimental period. Compared with CONT, the muscle weight and fiber cross-sectional area of the soleus were higher in STIM. The expression of muscle degradation markers (atrogin-1 and MuRF-1) in the soleus and EDL was lower in the STIM group than that in the CONT group. In contrast, B-SES did not significantly affect the expression of muscle synthesis (Eif4e and p70S6K) and mitochondrial (PGC-1α) markers. B-SES prevents muscle atrophy in CIA rats by reducing muscle degradation without exacerbating arthritis, demonstrating its promising potential as an intervention for RA-induced muscle atrophy.

## 1. Introduction

Rheumatoid arthritis (RA) is an autoimmune disease that primarily targets the synovial membranes, leading to arthritis and joint destruction [1,2]. RA also triggers muscle atrophy due to inflammation, leading to a decline in daily living activities [3,4]. Thus, preventing RA-induced muscle atrophy is an important aspect of RA treatment. Clinical reviews have strongly recommended exercise therapy for patients with RA [5]. However, getting enough exercise is occasionally difficult for these patients because of their arthritis and muscle atrophy. Therefore, developing alternative therapies for exercise is necessary to prevent RA-induced muscle atrophy.

Electrical muscle stimulation (EMS) induces automatic muscle contractions. Several studies have reported its preventive effects against muscle atrophy at the muscle fiber and molecular levels [6,7]. Conventional pad-type EMS delivers a band-shaped current that induces electrical stimulation on superficial muscles. Therefore, the efficiency of skeletal muscle contraction in the deep layers has been a concern in conventional EMS. Belt electrode-skeletal muscle electrical stimulation (B-SES) is a new type of EMS that makes use of belt-type electrodes; the entire surface functions as an electrode. B-SES utilizes tubular current and can induce effective electrical stimulation of multiple muscles [8,9]. In clinical practice, this innovative method is used to increase skeletal muscle mass in patients who have difficulty exercising, such as those undergoing hemodialysis [10] and patients with heart failure [11]. However, evidence regarding the efficacy of B-SES for RA-induced muscle atrophy remains insufficient.

Myofibers are classified as type I (slow-twitch muscle) and type II (fast-twitch muscle), which are based on the content of myosin heavy chain isoforms [12] and differ in fatigue resistance and contraction speed [13]. Collagen-induced arthritis (CIA) rats exhibit a pathology similar to patients with RA, including the progression of joint swelling and muscle atrophy due to inflammation [14,15]. The soleus is known to be rich in slow-twitch muscle fibers, whereas the extensor digitorum longus (EDL) is known to be rich in fast-twitch muscle fibers [16,17]. We have previously found that CIA rats predominantly exhibit atrophy in the soleus, which could be prevented via treadmill exercise.

We hypothesized that the effects of B-SES on the skeletal muscle of CIA rats depend on the muscle subtype, similar to the effects of treadmill exercise. In this study, we analyzed the soleus and EDL at the histological and molecular levels to investigate the effects of B-SES on muscle atrophy in CIA rats.

## 2. Results

### 2.1. Chronological Changes in Body Weight and Paw Volume

Day 0 was defined as the day of immunization, and physical changes in CIA rats were evaluated every other day, starting from day 0 to 28. We found that the body weight of B-SES-treated (n = 18) and untreated (n = 10) CIA rats decreased during days 10–12. However, body weight did not significantly differ between the B-SES-treated and untreated CIA rats throughout the entire period (Figure 1A).

B-SES was applied only to the right hindlimb. The hindlimbs of CIA rats were categorized into three groups: stimulation (STIM) group (n = 18), which included the right hindlimbs of B-SES-treated CIA rats; contralateral (CL) group (n = 18), which included the left hindlimbs of B-SES-treated CIA rats; and control (CONT) group (n = 20), which included both hindlimbs of untreated CIA rats. The paw volume increased in all groups from day 10, reaching a maximum between days 18 and 22, after which it gradually decreased. Notably, the paw volume in the STIM group was lower than that in the CONT group on day 18. Similarly, the paw volume in the CL group was lower than that in the CONT group on days 16 and 18 (Figure 1B).

### 2.2. Effect of B-SES on Relative Muscle Weight

The relative weights (normalized to body weight [mg/g]) of the soleus and EDL muscles are shown in Figure 2. The relative weight of the soleus muscle in the STIM group was significantly higher than that in the CONT group (*p* < 0.05), whereas there was no significant difference between the STIM and CL groups (CONT: 0.347 ± 0.055, STIM: 0.444 ± 0.080, CL: 0.390 ± 0.080 [mg/g]). In contrast, we found no significant difference in the relative weight of the EDL muscle among the three groups (CONT: 0.387 ± 0.046, STIM: 0.425 ± 0.086, CL: 0.400 ± 0.094 [mg/g]).

### 2.3. Analysis of the Muscle Fiber Cross-Sectional Area (CSA) in the Soleus and EDL

We histologically evaluated the soleus and EDL muscles in each group (n = 8 each) using picrosirius red staining. We examined the average muscle fiber CSA (Figure 3B,C) and its frequency distribution in the soleus and EDL (Figure 3D,E). The average muscle fiber CSA in the soleus in the STIM group was significantly higher than that in the CONT group (*p* < 0.05) but did not significantly differ compared with that in the CL group (CONT: 893 ± 54, STIM: 1037 ± 88, CL: 986 ± 121 [µm^2^]). Regarding the frequency distribution of the size of muscle fiber CSA in the soleus, the STIM group exhibited a lower frequency in the range of 200–800 µm^2^ than the CONT group.

In contrast, the average muscle fiber CSA in the EDL did not significantly differ among the three groups (CONT: 966 ± 47, STIM: 1050 ± 67, CL: 1013 ± 79 [µm^2^]). Regarding the frequency distribution of the size of muscle fiber CSA in the EDL, the STIM group exhibited a lower frequency in the range of 200–600 µm^2^ than the CONT group.

### 2.4. mRNA Expression in the Soleus and EDL on Day 28

mRNA expression in the soleus and EDL muscles in each group (n = 10 each) was evaluated 3 h after the last stimulation (B-SES or sham stimulation) on day 28 using real-time (RT)-PCR (Figure 4). We found that atrogin-1 and muscle RING-finger protein-1 (MuRF-1) mRNA expression in the soleus and EDL decreased in the STIM group compared with that in the CONT group (soleus: 0.51- and 0.56-fold, EDL: 0.61- and 0.55-fold, respectively, both *p* < 0.05). Meanwhile, the mRNA expression of atrogin-1 and MuRF-1 in the soleus decreased in the CL group compared with that in the CONT group (0.73- and 0.72-fold, respectively, *p* < 0.05). No significant differences in the mRNA expression of eukaryotic translation initiation factor 4E (Eif4e), 70 kDa ribosomal protein S6 kinase (p70S6K), and proliferator-activated receptor gamma coactivator 1-alpha (PGC-1α) in the soleus and EDL were observed among the three groups.

## 3. Discussion

In this study, B-SES increased muscle weight and muscle fiber CSA in the soleus but not in the EDL of CIA rats. At the molecular level, B-SES suppressed the mRNA expression of muscle degradation markers in the soleus and EDL of CIA rats but did not affect that of muscle synthesis and mitochondrial activity markers. These findings indicate that B-SES stimulation prevented muscle atrophy in CIA rats by inhibiting muscle degradation.

In our previous study, we observed progressive muscle atrophy in the soleus of CIA rats [18]. In this study, B-SES prevented this muscle atrophy in the soleus. We estimated that B-SES suppressed the decrease in muscle weight of the soleus by preventing the characteristic muscle atrophy in CIA rats. Honda et al. reported that B-SES increases the CSA of slow-twitch muscle fibers in immobilized rats but has no significant effect on fast-twitch muscle fibers [19]. The soleus has a higher proportion of slow-twitch muscle fibers than the EDL [16,17]. We hypothesized that the differences in muscle fiber composition may have influenced the differential effects of B-SES in preventing atrophy of the soleus and EDL.

At the molecular level, skeletal muscle mass is regulated by muscle degradation, muscle synthesis, and mitochondrial activity [20,21]. The expression of muscle degradation markers, such as atrogin-1 and MuRF-1, has been reported to increase in CIA rats [14,15]. Previous studies have shown that B-SES or electrical stimulation suppresses the expression of muscle degradation markers in the soleus [7,22], tibialis anterior [23], and gastrocnemius [23,24] of immobilized model rats. Consistent with these results, we found a decrease in the mRNA expression of muscle degradation markers in the soleus.

The effects of B-SES on muscle synthesis and mitochondrial activity have been controversial. Several studies on B-SES or electrical stimulation using immobilized rats have shown an increase in the expression levels of muscle synthesis markers via activation of the mTOR pathway [23,25], whereas others have reported no significant changes [7] or that repeated electrical stimulation decreased the protein expression level of p70S6K [21]. Similarly, a previous study has reported that B-SES increases the mitochondrial activity of immobilized rats at the mRNA level [22], whereas another study reported no significant changes at the protein level [23]. In this study, no significant changes were observed in the mRNA expression of muscle synthesis and mitochondrial activity markers in the soleus and EDL of CIA rats. Unlike muscle degradation, muscle synthesis and mitochondrial activity were not significantly affected in CIA rats; thus, the B-SES-mediated prevention of muscle atrophy in CIA rats may be mainly attributed to the inhibition of muscle degradation.

Interestingly, we found that compared with the control group, the expression of muscle degradation markers was reduced in the soleus of the contralateral hindlimb, which did not directly receive B-SES. Previous studies have also reported that electrical stimulation of one hindlimb suppresses the expression of muscle degradation markers in the contralateral hindlimb of immobilized rats [7]. Thus, applying B-SES to one hindlimb affects the contralateral hindlimb through systemic factors, such as myokines and the central nervous system. Future studies are needed to reveal the systemic effects of B-SES.

This study confirmed that B-SES did not increase paw volume in CIA rats. Previous studies have reported that electrical stimulation suppresses the expression of inflammatory cytokines in the plasma of adjuvant-induced arthritis rats, preventing an increase in paw volume [26]. The absence of arthritic exacerbations is important for clinical applications. With the advancement of drug therapies over the past few decades, the percentage of patients achieving remission has increased [27]. However, prevention and treatment of RA-induced muscle atrophy remain difficult [28]. This study provides new insights into the prevention of muscle atrophy in patients with RA. B-SES may enable patients with RA who have difficulty exercising to maintain their daily living activities without exacerbating arthritis.

Despite the findings, this study had some limitations. First, whether the electrical stimulation protocol used was the most effective intervention was unclear, as only single B-SES protocols were implemented in this study. Second, molecular level and tissue analysis data were not available for all experimental time points. The expression of muscle degradation, muscle synthesis, and mitochondrial activity markers may have differed depending on the timing of muscle sampling and analysis. However, the results of the molecular analysis were consistent with those of the histological analysis. Lastly, the pathway through which B-SES prevented muscle atrophy in CIA rats was partially elucidated. Further investigation is needed to clarify the effects of B-SES on synovitis and its systemic effects.

## 4. Materials and Methods

### 4.1. Animals

A total of 28 8-week-old male Dark Agouti (DA) rats (body weight, 155–178 g; Shimizu Laboratory Suppliers, Kyoto, Japan) were used in this study. The rats were housed under a 12 h light/dark cycle and were provided with free access to food and water.

The experimental protocol was approved by the Ethics Review Committee for Animal Experimentation of Kyoto Prefectural University of Medicine (approval no. M2024-261). All experimental procedures were performed under anesthesia, and all efforts were made to minimize animal suffering.

### 4.2. Experimental Design

#### 4.2.1. Collagen-Induced Arthritis Model

Type II collagen (CII; Collagen Research Center, Tokyo, Japan) and Freund’s incomplete adjuvant (FIA; Sigma-Aldrich, St. Louis, MO, USA) were mixed and emulsified on ice in a 1:1 ratio. The CII/FIA solution (200 μL) was then intradermally injected into the base of the tail of the rats [29]. Day 0 was defined as the day of immunization with the CII/FIA solution.

#### 4.2.2. B-SES Protocol

B-SES was performed using an electrical stimulator designed for small animals (Homer Ion Corp., Tokyo, Japan), which comprised a B-SES main unit and belt-type electrodes. To ensure effective electrical stimulation, body hair on the proximal and distal parts of the hindlimb was shaved, and the belt-type electrodes were wrapped around them. The experimental B-SES condition was as previously reported [22]: a stimulus frequency of 50 Hz, stimulus intensity of 4.7 mA, stimulus cycle of 2 s stimulation/2 s off, and stimulation time of 15 min. B-SES was applied a total of four times from days 14 to 28 (Figure 5). B-SES was performed under mixed anesthesia via intraperitoneal administration of three different types of anesthetic drugs (0.375 mg/kg medetomidine [Nippon Zenyaku Kogyo Co., Ltd., Fukushima, Japan], 2.0 mg/kg midazolam [Sandoz Pharma Co., Ltd., Tokyo, Japan], and 2.5 mg/kg butorphanol [Meiji Seika Pharma Co., Ltd., Tokyo, Japan]).

#### 4.2.3. Experimental Group of CIA Rats

The 28 CIA rats were randomly divided into B-SES-treated and untreated (no B-SES) rats (n = 18 and 10, respectively). The 18 B-SES-treated CIA rats received B-SES only in the right hindlimb, whereas the 10 untreated CIA rats received anesthesia but not electrical stimulation (sham stimulation). Accordingly, the hindlimbs of CIA rats were categorized into three groups: B-SES stimulation (STIM) group, which included the right hindlimbs of B-SES-treated CIA rats that received B-SES; B-SES contralateral (CL) group, which included the left hindlimbs of B-SES-treated CIA rats that did not receive B-SES; and control (CONT) group, which included both hindlimbs of untreated CIA rats (no B-SES). All rats were humanely euthanized 3 h after the last B-SES or sham stimulation on day 28. In each group, 10 hindlimbs were used for molecular analyses, while 8 were used for histological analysis (Figure 6).

### 4.3. Body Weight and Paw Volume

Body weight and paw volume were measured every other day from days 0 to 28. Paw volume was measured using a water replacement plethysmometer (Unicom Japan, Tokyo, Japan) [18].

### 4.4. Muscle Preparation

After euthanization on day 28, the soleus and EDL in all groups were collected and weighed. Standardized dissection methods were employed to remove excess fat and connective tissue. For histological analysis, the muscles were fixed in 4% paraformaldehyde (Wako, Osaka, Japan). For transcriptional analysis, the muscles were incubated in RNA Protect^®^ Tissue Reagent (QIAGEN, Hilden, Germany) and stored at 4 °C until analysis.

### 4.5. Histological Analysis

The muscles were embedded in paraffin, and transverse 10 µm-thick sections were obtained from the central region of the muscle using a cryostat. The muscle cross-sections were stained with picrosirius red, as previously described [18]. Representative images of the soleus and EDL muscles were captured using a microscope at 20× magnification. The muscle fiber CSA ranging from 200 to 3000 µm^2^ was automatically measured from the entire cross-section of the soleus and EDL using a BZ-X700 BZ analyzer (Keyence, Osaka, Japan). The average muscle fiber CSA and frequency distribution of muscle fiber CSA of the soleus and EDL were compared among the three groups.

### 4.6. RT-PCR

The soleus and EDL muscles were snap-frozen in liquid nitrogen and subsequently pulverized using a Cryo-Press CP-100WP (Microtech Nichion, Chiba, Japan). The pulverized muscle tissue was collected, and total RNA was isolated using ISOGEN II (NIPPON Gene, Osaka, Japan) and reverse transcribed to cDNA using a ReverTra Ace^®^ qPCR RT Master Mix (TOYOBO, Osaka, Japan) according to the manufacturer’s protocol. Fast quantitative RT-PCR was conducted on a Step One Plus system (Applied Biosystems) using predesigned TaqMan probes (Applied Biosystems, Foster City, CA, USA). The markers used were as follows: atrogin-1 (Rn00591730_m1) and MuRF-1 (Rn00590197_m1) as muscle degradation markers; p70S6K (Rn00583148_m1) and Eif4e (Rn00821567_g1) as muscle synthesis markers; and PGC-1α (Rn00580241_m1) as a mitochondrial activity marker. Each reaction had a total volume of 20 µL containing 2 µL of cDNA and 10 µL of the KAPA Probe Fast ABI Prism qPCR kit (KAPA Biosystems, Cape Town, South Africa) with specific primers for each target gene. The PCR amplification protocol included 40 cycles of denaturation at 95 °C for 15 s, followed by annealing and extension at 60 °C for 1 min. Relative changes in gene expression were calculated using the comparative ΔΔCq method [30], with 18S ribosomal RNA as the internal control. The sequences of the primers used for the amplification of the 18S ribosomal RNA were as follows: forward primer, 5′-ATGAGTCCACTTTAAATCCTTTAACGA-3′; reverse primer, 5′-CTTTAATATACGCTATTGGAGCTGGAA-3′; and probes, 5′-(FAM)ATCCATTGGAGGGCAAGTCTGGTGC(BHQ)-3′.

### 4.7. Statistical Analysis

Statistical analysis was performed using EZR software (version 1.68, Saitama Medical Center, Jichi Medical University; Saitama, Japan) [31]. The results are presented as the mean ± standard deviation (SD). Statistical differences were evaluated using an unpaired *t*-test for two-group comparisons and parametric one-way analysis of variance (ANOVA) for multiple-group comparisons. Bonferroni’s post-hoc test was used to determine the specific differences between groups if the results were significant. Statistical significance was set at *p* < 0.05. Graphical representations of the results were generated using GraphPad Prism software (Version 10.1.1) (GraphPad Software, San Diego, CA, USA).

## 5. Conclusions

B-SES inhibited muscle atrophy in the soleus of CIA rats by suppressing muscle degradation without exacerbating arthritis. The findings of this study suggest that B-SES is an effective intervention for preventing RA-induced muscle atrophy.

## Figures and Tables

**Figure 1 ijms-26-03294-f001:**
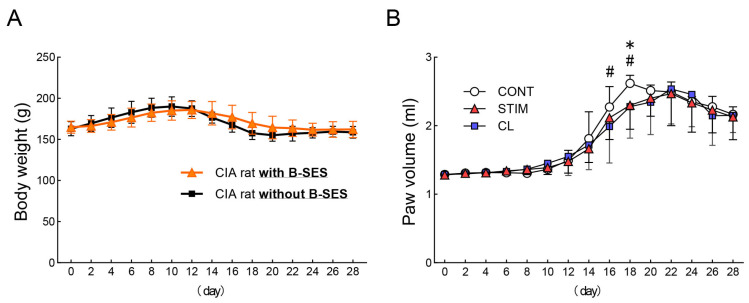
Physical changes in CIA rats. Body weight (**A**) and paw volume (**B**) were measured every other day from days 0 to 28. Data are expressed as the mean ± standard deviation. (**A**) B-SES-treated CIA rats: n = 18, untreated CIA rats: n = 10. (B) * CONT vs. STIM, # CONT vs. CL (*p* < 0.05). CONT: n = 20, STIM: n = 18, CL: n = 18. CONT, control; STIM, stimulation; CL, contralateral.

**Figure 2 ijms-26-03294-f002:**
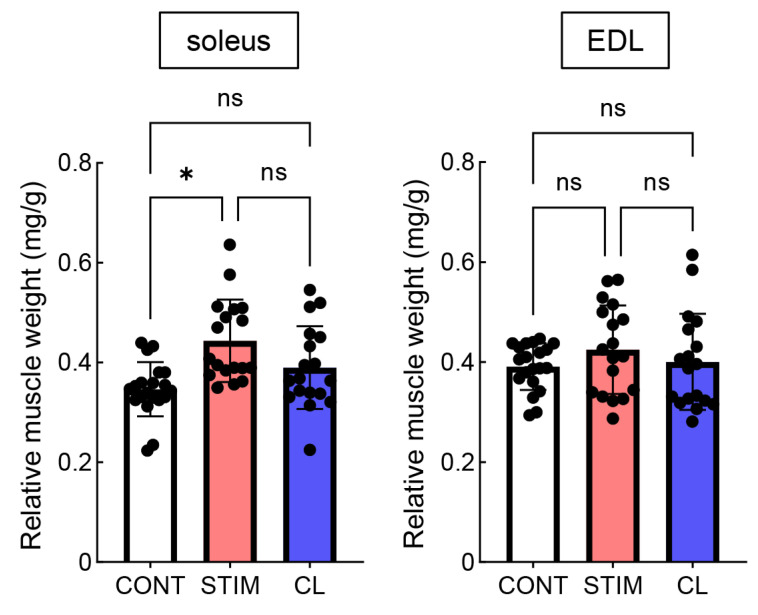
Relative weight of the soleus and EDL. Soleus and EDL muscle weights were normalized to body weight (mg/g). White, red, and blue bars represent the CONT, STIM, and CL groups, respectively. Black dots represent data from each hindlimb. Data are expressed as the mean ± standard deviation. ns, no significant difference. * *p* < 0.05. CONT: n = 20, STIM: n = 18, CL: n = 18. CONT, control; STIM, stimulation; CL, contralateral.

**Figure 3 ijms-26-03294-f003:**
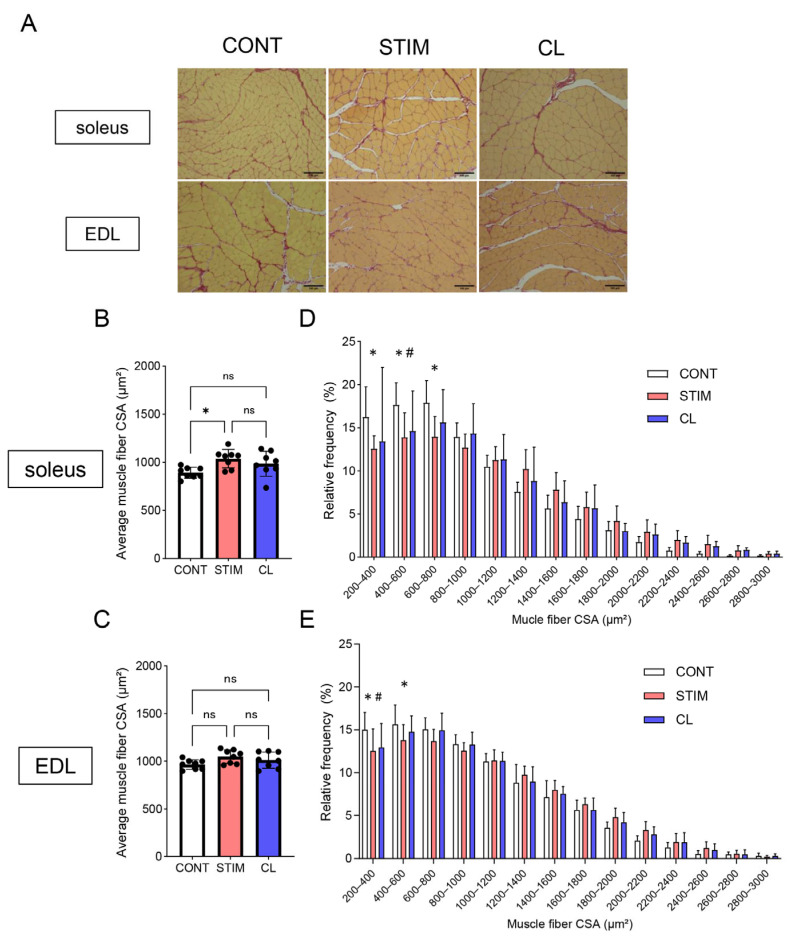
Histological analysis of the soleus and EDL. (**A**) Representative microscopic images of picrosirius red staining in the soleus and EDL. Scale bar, 100 μm. (**B**,**C**) Average muscle fiber CSA in the soleus (**B**) and EDL (**C**). White, red, and blue bars represent the CONT, STIM, and CL groups, respectively. Each black dot represents a sample. Data are expressed as the mean ± standard deviation. CSA, cross-sectional area; ns, no significant difference. * *p* < 0.05. n = 8 per group. (**D**,**E**) Frequency distribution of muscle fiber CSA in the soleus (**D**) and EDL (**E**). The vertical axis shows the frequency distribution of muscle fiber CSA (%), while the horizontal axis shows the size of the muscle fiber CSA (μm^2^). * CONT vs. STIM, # CONT vs. CL (*p* < 0.05). n = 8 per group. CONT, control; STIM, stimulation; CL, contralateral.

**Figure 4 ijms-26-03294-f004:**
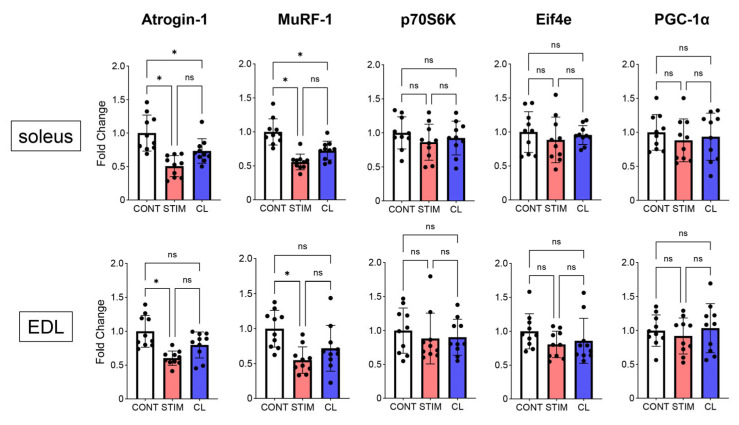
mRNA expression in the soleus and EDL. mRNA levels of atrogin-1, MuRF-1, p70S6K, Eif4e, and PGC-1α in the soleus and EDL were analyzed. White, red, and blue bars represent the CONT, STIM, and CL groups, respectively. Each black dot represents a sample. Data are expressed as the mean ± standard deviation. MuRF-1, muscle RING-finger protein-1; Eif4e, eukaryotic translation initiation factor 4E; p70S6K, the 70 kDa ribosomal protein S6 kinase; PGC-1α, proliferator-activated receptor gamma coactivator 1-alpha; ns, no significant difference. * *p* < 0.05. n = 10 per group. CONT, control; STIM, stimulation; CL, contralateral.

**Figure 5 ijms-26-03294-f005:**
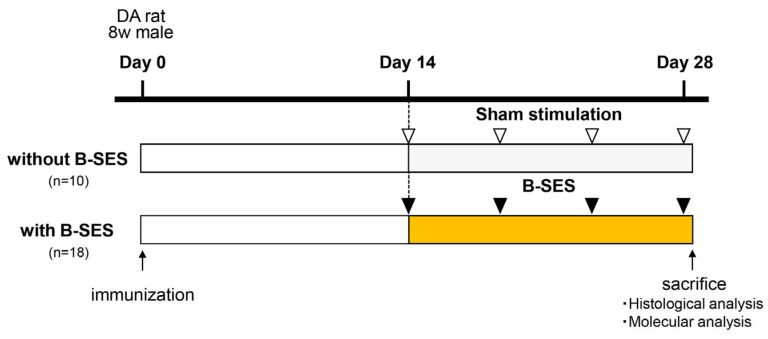
B-SES protocol. DA rats were immunized with a mixture of type II collagen and Freund’s incomplete adjuvant (day 0). B-SES or sham stimulation was applied four times from days 14 to 28. CIA rats were euthanized 3 h after the last stimulation on day 28. White triangles indicate sham stimulation, and black triangles indicate B-SES. CIA, collagen-induced arthritis; DA, Dark Agouti; CONT, control; STIM, stimulation; CL, contralateral; B-SES, belt electrode-skeletal muscle electrical stimulation.

**Figure 6 ijms-26-03294-f006:**
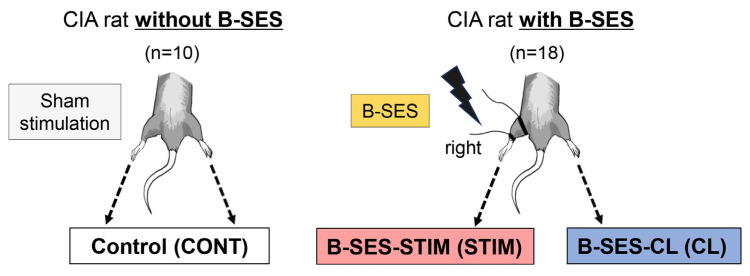
Animal model and research design. Twenty-eight CIA rats were randomly divided into B-SES-treated (n = 18) and untreated (n = 10) CIA rats. The groups were as follows: STIM group, which comprised the electrically stimulated right hindlimbs of B-SES-treated CIA rats; CL group, which comprised the non-stimulated left hindlimbs of B-SES-treated CIA rats; and CONT group, which comprised both hindlimbs of untreated CIA rats. CONT, control; STIM, stimulation; CL, contralateral; B-SES, belt electrode-skeletal muscle electrical stimulation.

## Data Availability

Data are contained within the article.

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
