# Peer review of "Belt Electrode-Skeletal Muscle Electrical Stimulation Prevents Muscle Atrophy in the Soleus of Collagen-Induced Arthritis Rats"

_ijms, 2025, doi:10.3390/ijms26073294_

Round 1

Reviewer 1 Report

Comments and Suggestions for Authors

Dear authors

I have two questions. Please answer to these questions in discussion. It will improve the quality of this paper.

  1. The electrical stimulation of Soleus and EDL significantly reduced the expression of Atrogin-1 and MuRF-1. This was reflected in significant increase of the relative weight of Soleus muscle, but not EDL muscle. Why? What is your hypothesis.
  2. You have found that: "compared with the control group, the expression of muscle degradation markers was reduced in the soleus of the contralateral hindlimb, which did not directly receive B-SES. Thus, applying B-SES to one hindlimb affects the contralateral hindlimb through systemic factors, such as myokines." How Is the central nervous system involved in muscle degradation process?< !--StartFragment -->

    < !--EndFragment -->

Author Response

We thank the reviewers for their helpful and constructive evaluation of our manuscript. We have revised the manuscript accordingly, and below are our responses and changes to address each of the comments.

The electrical stimulation of Soleus and EDL significantly reduced the expression of Atrogin-1 and MuRF-1. This was reflected in significant increase of the relative weight of Soleus muscle, but not EDL muscle. Why? What is your hypothesis.

Response/Action:

In this study, although B-SES suppressed the expression of Atrogin-1 and MuRF-1 in both the soleus and EDL, a significant increase in the relative weight was found only in the soleus muscle. In our previous study, we confirmed that in CIA rats, muscle atrophy progressed more prominently in the soleus compared to the EDL (Kamada Y, 2021). We assumed that muscle degradation was suppressed in both the soleus and the EDL, but a significant difference in skeletal muscle mass was observed only in the soleus muscle, which had undergone muscle atrophy due to CIA. We added the following sentence to the Discussion section (Line 156):

We estimated that B-SES suppressed the decrease in muscle weight of the soleus by preventing the characteristic muscle atrophy in CIA rats.

You have found that: "compared with the control group, the expression of muscle degradation markers was reduced in the soleus of the contralateral hindlimb, which did not directly receive B-SES. Thus, applying B-SES to one hindlimb affects the contralateral hindlimb through systemic factors, such as myokines." How Is the central nervous system involved in muscle degradation process?

Response/Action:

We appreciate the reviewer’s opinion. We also think that the nervous system controls skeletal muscles and can produce systemic effects. We added this point to the Discussion part (Line 188) as follows:

Thus, applying B-SES to one hindlimb affects the contralateral hindlimb through systemic factors, such as myokines and central nervous system.

Reviewer 2 Report

Comments and Suggestions for Authors

The manuscript investigated the effects of belt electrode-skeletal muscle electrical stimulation (B-SES) on muscle atrophy in collagen-induced arthritis (CIA) rats. The study is well-conceived and addresses an important issue in rheumatoid arthritis management. The clarity of data presentation and the relevance of the research question are commendable. The manuscript may warrant publication following minor revisions.

1. B-SES protocol

Please provide a detailed rationale for the chosen stimulation parameters. Specifically, address the key parameters explaining why these were selected based on existing literature or preliminary studies.

2. Experimental group of CIA rats:
Please provide a detailed rationale for the chosen stimulation parameters. Specifically, address the key parameters such as frequency, intensity, and duration, explaining why these were selected based on existing literature or preliminary studies.

3. Results:

In Figure 1B, how to explain the evident drop in paw volume for the CONT group between days 18-28?

Moreover, from day 20 onwards, the paw volume shows no significant differences between the CONT group and the STIM/CL groups. Does this suggest that the efficacy of B-SES diminishes in the later stages of treatment?

4. Discussion:

Is monitoring muscle condition sufficient for evaluating the overall effectiveness of B-SES treatment?

Also, please elaborate on whether muscle improvements reflect the broader treatment outcomes for CIA or RA.

Author Response

We thank the reviewers for their helpful and constructive evaluation of our manuscript. We have revised the manuscript accordingly, and below are our responses and changes to address each of the comments.

  1. B-SES protocol

Please provide a detailed rationale for the chosen stimulation parameters. Specifically, address the key parameters explaining why these were selected based on existing literature or preliminary studies.

Response/Action:

Takahashi A et al. reported that electrical stimulation at 4.7 mA effectively prevents muscle atrophy by inducing 60% of maximal voluntary contraction in the hindlimb muscles of rats. In their study, they adopted a B-SES protocol (50 Hz, 2 seconds of stimulation followed by 2 seconds of rest, 15 minutes per session) and observed its inhibitory effect on soleus muscle atrophy in rats (Takahashi A, 2024). Based on this paper, we decided our B-SES protocol. We have added the study by Takahashi A et al. as a reference in the B-SES protocol section of our manuscript (Line 232).

  1. Experimental group of CIA rats:

Please provide a detailed rationale for the chosen stimulation parameters. Specifically, address the key parameters such as frequency, intensity, and duration, explaining why these were selected based on existing literature or preliminary studies.

Response:

Takahashi A et al. applied B-SES for two weeks and observed its inhibitory effect on soleus muscle atrophy in rats (Takahashi A. 2024). In another study, Uno H et al. applied B-SES (60Hz, 3 mA) for one week, and observed its inhibitory effect on tibial anterior and gastrocnemius muscle atrophy in rats (Uno H. 2024). Based on these studies, we set the B-SES application duration as two weeks in this study, expecting an increase in muscle mass in CIA rats.

  1. Results:

In Figure 1B, how to explain the evident drop in paw volume for the CONT group between days 18-28?

Response:

In our previous study using the CIA rat model, a decrease in paw volume was observed after 18 days of immunization (Kamada Y, 2021). In this study, the course of paw volume changes is similar to that in our previous study, supporting the validity of the results.

Moreover, from day 20 onwards, the paw volume shows no significant differences between the CONT group and the STIM/CL groups. Does this suggest that the efficacy of B-SES diminishes in the later stages of treatment?

Response/Action:

As the reviewer pointed out, the absence of significant differences after day 20 may suggest a plateau in B-SES efficacy on arthritis. We evaluated only paw volume and did not perform a detailed histological analysis of synovitis. Therefore, while B-SES did not exacerbate arthritis, we could not conclude that it ameliorated arthritis. Additionally, this study mainly focused on the effect of B-SES on skeletal muscle. Therefore, we have decided not to make any changes to the following sentence in the Discussion section (Line 191):

This study confirmed that B-SES did not increase paw volume in CIA rats.

  1. Discussion:

Is monitoring muscle condition sufficient for evaluating the overall effectiveness of B-SES treatment?

Response/Action:

This study was conducted regarding the effects of B-SES on the skeletal muscles of CIA rats. This study confirmed that B-SES inhibited the reduction of soleus muscle weight and myofiber cross-sectional area via suppression of muscle degradation marker. However, the effects of B-SES on synovitis and its systemic effects still need to be further investigated.

We have revised the final Limitation (Line 207) to the following sentence:

Lastly, the pathway through which B-SES prevented muscle atrophy in CIA rats was partially elucidated. Further investigation is needed to clarify the effects of B-SES on synovitis and its systemic effects.

Also, please elaborate on whether muscle improvements reflect the broader treatment outcomes for CIA or RA.

Response/Action:

Muscle improvement leads to many clinical benefits for patients with rheumatoid arthritis. Improved muscle strength and endurance are associated with enhanced activities of daily living, reducing the need for assistance and enabling patients to live independently. Additionally, muscle improvement also contributes to improvement in the quality of life for RA patients. We have already described the benefits of these muscle improvements for RA patients in the Discussion section and therefore did not make any changes to our manuscript (Line 197).

Round 2

Reviewer 1 Report

Comments and Suggestions for Authors

Dear authors, I am satisfied with your responses and the revision of the article.